# Navigating Purchase Intentions: The Influence of Reviewers’ Comments Moderated by Risk and Trust

**DOI:** 10.3390/bs14070552

**Published:** 2024-06-28

**Authors:** Sabina Kwakye, Ahmet Ertugan, Laith Tashtoush

**Affiliations:** 1Business Administration Department, Near East University, 99138 Nicosia, Cyprus; laith.tashtoush@neu.edu.tr; 2Marketing Department, Near East University, 99138 Nicosia, Cyprus; ahmet.ertugan@neu.edu.tr

**Keywords:** reviewers’ comments, social media, purchase intention, risk, trust

## Abstract

Despite previous research on the impact of social media reviews on purchase intention, it is still unclear how individual reviewers’ comments impact purchase intentions. To fill in the gap, this study examines the impact of product-related comments on purchase intentions. In terms of contributions, this study evaluates people’s dependence on social media for product information and purchase intention while considering risk and trust as moderating factors. The data were collected quantitatively using surveys. The sample consisted of 384 academically advanced adults with high social media engagement. The study hypotheses were tested using the PROCESS macro after exploratory and confirmatory factor analysis, and correlation analysis were conducted. Among those surveyed, it was confirmed that most people sought product-related comments on social media when seeking information about a product they might be interested in purchasing. It was also observed that the risk factor reduced the impact of other variables in the model presented in this study. In conclusion, this investigation is more reliable, and its outcomes benefit scholars, supervisors, merchants, and webmasters, for instance, in establishing a study for future research efforts and providing practical guidance that could boost promotional efforts and marketing activities, especially in this era of artificial intelligence.

## 1. Introduction

Social media has influenced significant aspects of human life [1]. With masses of people present on social media, it has become an “online world” in which people live [2,3]. This new life on social media has become so natural that most people make significant decisions based on the information they receive or see on social media, including judgments on purchases [4]. Therefore, social media is now considered a significant factor influencing online purchase decisions [5].

Purchase intention is one of the final stages before an individual makes a purchase, and it occurs when an individual decides to patronize a product or service and is willing to part ways with their money in exchange for the product or service [6]. In deciding to purchase, several factors come into play. Sometimes, people do not even know that they need a product or service until they are exposed to information about it [7]. This is because social media exposes users to information that affects their willingness to purchase and facilitates their decision-making process [5,8].

Most marketers use social media as a tool to drive customers’ purchase intentions [9] through paid advertisements, as well as unpaid or organic advertisements. Reference [6] explained that social media has become a means of advertising to target populations and influence purchase intentions. Most organizations also use social media influencers to advertise their products and share information that they intend to spread to their existing and potential customers [10]. The justification for organizations’ use of social media influencers and celebrities in advertisements is that the followers of these celebrities and media influencers tend to relate more to them, and as such, they believe that these celebrities and media influencers are trustworthy [11]. As [12] advanced, credibility is the most important factor regarding online advertisements.

Marketers are mostly concerned about the perceptions of their existing and target customers related to the credibility of the messages, information, and advertisements they share [12]. Recently, marketers have recognized the importance of customer reviews and word-of-mouth referrals in improving the credibility of their products, services, or brands. Several studies have investigated the usefulness of online reviews in influencing purchase intentions, especially when those reviews are considered trustworthy or credible [13,14]. One study that addressed the usefulness of online reviews in influencing purchase intentions was [15]: it concluded that hotels received a greater number of bookings when there were existing and positive online reviews about the hotel.

With the constant increase in technology usage, it is not unexpected that most people seek information about products and services through social media rather than traditional media such as [5,6,8]. Existing studies have shown that as people constantly seek these products and services on social media, they develop a dependency on social media [5,16,17,18]. In other words, constantly seeking product and service information on social media leads to dependency on social media for product information; this dependency on social media affects purchase intentions [6,8,18].

The current study intends to examine how risks and trust moderate the relationship between dependency on social media for reviewers’ comments and purchase intentions. This research article makes several contributions. It is the first study to focus on the subtle consequences of social media reviewer comments on purchase intention. The majority of research that is presently available aggregates all social media interactions without differentiating between the unique effects of perceived risk and trustworthiness related to “individual reviewer comments”. Previous research addressed social media reviews as a concept, and how social media reviews influence purchase intention, with little research addressing the impact of risk and trust in the relationship between social media reviews and purchase intention. There is, however, a gap in the research regarding the subtle influence of individual reviewers’ comments as opposed to general social media reviews. As such, this study aims to fill in the gap by specifically addressing individual reviewer’s comments. Additionally, the focus of earlier research on social media dependence was on people’s addictions to platforms rather than their reliance on social media for product information. This study addressed social media dependence as the reliance on social media, rather than an addiction to social media. In addition, this study was conducted with diverse academically advanced adults with high social media engagement, from different countries and continents of the world, stationed in a region (North Cyprus) that has not been explored in previous research. This study, therefore, provides information on the social media influence dynamics based on responses from adults that have knowledge and exposure to different cultural perceptions and social media content. Furthermore, a varied participant pool improves the generalizability of the results since it provides less skewed responses, hence, improving the understanding of how risk and trust are viewed in relation to social media-driven purchase intentions. In view of this, the research question guiding this study is as follows: do trust and perceived risk of reviewer comments on social media moderate the relationship between consumers’ dependence on social media for product information and their subsequent purchase intentions?

### 1.1. Literature Review

#### 1.1.1. Seeking Product-Related Reviews on Social Media and Purchase Intentions

The opinion or feedback of customers regarding a particular product or brand constitutes a product or brand review [19]. A growing number of businesses are advertising their products and services through social media [20]. It is rare for internet users to read web pages in full detail. Instead, they scan the page and retrieve only the needed information [14,21]. Searching for information should be quick and easy, and people should put little effort into finding what they need [14]. As social networking media elements have gained popularity and improved, customers now have more options for accumulating product-specific information, which provides numerous avenues for consumers to contribute their consumption-related suggestions through engagement with online communities in the form of reviews [22]. As such, a reviewer is any person who delivers feedback, criticism, or information about a product or service. 

The growth of online consumer reviews has made them an important marketing communication tool because many consumers use them as the first step in their shopping process. Ratings and reviews have a considerable impact on social presence, proximity, familiarity, and informational support, but they do not significantly affect emotional support or purchase intentions, according to a study by [23] concerning ratings and reviews. This, however, runs counter to the conclusions made by [5].

People connect with content on social media by watching, reading, or hearing it and creating their own content [24]. As a result, it is an excellent resource for information for many people. Social media has been hailed as one of the most essential sources for purchasing and influencing consumer decision making [23]. Most individuals actively look for product information on social media.

Online customer reviews simplify customer decision making by reducing customer mental strain while simultaneously increasing purchase intention [25]. The decision-making process is also heavily influenced by the nature of the product and the traits of the users. Compared with products with negative reviews on electronic Word of Mouth (eWOM) platforms, positive reviews receive more recommendations from friends. Instead of depending only on advertisements, buyers frequently read prior customer reviews before deciding what to buy [25]. Consumers can gain more confidence in their purchase decisions by reviewing online comments. 

The impact of product reviews on purchase intentions on social media can be explained by the social influence theory. The social influence theory asserts that people’s attitudes, beliefs, and behaviors are frequently influenced by their social environment and the ideas of those around them [26]. When consumers search social media for reviewer’s comments, they are essentially seeking social proof, which is the psychological phenomenon in which people believe that other people’s actions indicate the right course of action in a particular circumstance. Purchase intentions can be influenced by positive or negative comments from reliable influencers or peers. Social influence can be normative or informational: the normative effect fosters a sense of conformity and belonging, whereas the informational influence of online reviews gives the impression that the information is more diagnostic and accurate [26,27]. Accordingly, customers’ tendency to acquire the goods is higher the more positively worded the online review is and vice versa [28]. Based on the reviewed literature and theory, the following hypothesis is proposed:
**H1:** *Seeking reviewers’ product-related comments on social media affects purchase intentions.*

#### 1.1.2. Dependency on Social Media for Reviewer’s Comments on Products

Dependency is defined by [29] as a relationship where one party depends on the resources of another to satisfy needs or to reach goals. The distinctive and essential conceptualization of media dependence relations is the foundation for MSD, which explains the reasons behind and effects of people’s media use [16]. This study examines how social media use can alter customers’ views about buying products using the well-known media system dependence theory (MSD) as a theoretical framework.

Media system dependency theory refers to the following phenomenon: According to [5], a dependence relationship is one in which the resources of one person assist the other in meeting their needs, hence preserving the relationship. Reference [16] reported that when people’s needs are met any time they interact with media, they become dependent on media. The media has a significant impact if people rely on them whenever they need to obtain a piece of information [16,30]. Audience dependency is determined by how long a medium is used [18]. In addition, an individual’s greater dependency on a medium may be due to the medium meeting their needs [16].

When one party needs another’s resources to achieve its goals, the party repeats the procedure or is likely to repeat the process each time it needs that resource, reinforcing dependency [31]. People frequently seek information on social media and depend on it for resources and information to accomplish a goal: decision making [16,32]. Therefore, using social media to research products and make decisions leads to dependency on social media [33].

Individuals are more prone to forming dependent connections with the media depending on how well it satisfies their expectations and wants, which affects the manner in which they utilize it [17]. According to [16], when social media gives people the information they need, they become utterly dependent on it. Based on their findings, [18] further confirmed that people’s time in the media determines their dependency. Additionally, the literature has demonstrated that using social media for informational purposes makes one dependent on it. Moreover, the literature has shown that people increasingly rely on social media to research products [5,16,17,18]. Based on this, the following hypotheses are proposed:
**H2:** *Seeking reviewers’ product-related comments has a positive impact on dependency on social media.*
**H3:** *Social media dependency mediates the relationship between seeking reviewers’ product-related comments and purchase intentions.*

#### 1.1.3. Risks of Buying a Product

Risk in the context of this research refers to the fear of a product not meeting the expectations or needs of the customer, a product not being sustainable, or a product that can cause reputational damage to the user. When a review of a product provides negative information about the product, consumers perceive the product to be risky [34]. There is ample evidence that perceived risk negatively impacts purchase intentions [15,35]. Reference [15] demonstrated that consumers’ perceptions of the possibility of product malfunction have a negative impact on their intention to purchase a product.

Media systems dependency theory accurately captures the impact of risk in the relationship between seeking reviewers’ product-related comments on social media and purchase intention. According to MSD theory, when it comes to purchasing intentions and social media reviews, high-risk purchases boost dependency: this is because customers are more uncertain and more likely to experience unfavorable consequences while making high-risk purchases. They are more likely to rely largely on social media comments for reliable and in-depth product information in an effort to reduce this danger. Because of this increased dependence, the knowledge gained from these reviewers’ comments is quite important in determining the things that they intend to buy. Customers are using social media more frequently in high-risk situations because of the interactive and social aspects of these platforms, which also enable them to ask other users for immediate guidance and comments.

Reference [5] states that people who utilize social media and believe that products with negative comments are detrimental are more likely to be aware of the negative aspects of the products and to adopt the negative opinions of other users about the products. Risk harms consumers and will likely reduce their purchase intentions [34]. If seeking product information on social media makes people dependent on social media and a reviewer’s comment that portrays the risk of a product possesses the ability to influence purchase intention, then the following hypothesis is proposed:
**H4:** *Risk moderates the relationship between dependency on social media reviewers’ comments and purchase intentions.*

#### 1.1.4. Trust in the Reviewer’s Comments

Trust in the context of this research refers to the belief that reviews about a product are trustworthy and that the product will meet the customer’s expectations. The findings of a study published by [7] suggest that customers depend on expert sources, popularity indications, and two-sided reviews to inform their choice of services. In other words, believing that a product or service’s performance and quality are the same as or comparable to those of prior users’ reviews is the basis for trusting product reviews on social media.

In online business transactions, trust is the most critical factor [23,36]. Recently, marketing managers have recognized that online reviews play an important role in the decision-making process of customers. Research indicates that online consumer reviews might be a more reliable information resource than information created by sellers [37]. Due to the growing impact of online consumer reviews (OCRs) on consumers’ decision-making processes, online vendors have started incorporating OCRs into their advertising. Compared with ads, reviews are more trustworthy in online shops. Purchase intentions are greater among potential consumers when OCRs are perceived as more trustworthy [38]. For this reason, trust in reviews influences purchase intentions [23,37].

Complementary models that explain how trust moderates the relationship between social media reviews and purchase intentions are provided by the media system dependency (MSD) theory and the social influence theory. Both ideas shed light on the processes by which trust affects how social media reviews affect consumers’ choices. When the information source can be trusted, social influence works better [38]. Trusted sources have a greater chance of being accepted and followed by customers, which increases their propensity to make purchases. The MSD theory states that people rely on media to meet a variety of requirements, and the extent of this dependence affects how the media affects their views and actions. Purchase intention can be influenced by one’s level of trust in the media source, including comments made by social media reviewers [38].

If people become dependent on social media as a result of seeking product information through social media and their trust in product information on social media affects their purchase intentions, then the following hypothesis is proposed:
**H5:** *Trust moderates the relationship between dependency on social media reviewers’ product-related comments and purchase intentions.*

#### 1.1.5. Dependency on Social Media and Purchase Intentions

It has been empirically demonstrated that media dependency positively influences purchase intentions because media provides sufficient product information. Consumers are hesitant to purchase products if they believe they lack the information necessary to make an informed decision [39]. The lack of specific product information or the difficulty in locating it through traditional offline communication channels confuses customers [39,40]. However, user-generated information offered by social media platforms can significantly impact consumers’ purchase decisions because it is more enriched and more accessible to retrieve [41].

Making informed decisions during a purchase is difficult for consumers who lack knowledge about a particular product [13]. As a result, consumers may delay or even stop making purchases to avoid cognitive strain, thus influencing their product choice. Social media benefits customers by offering thorough information from several sources, making them dependent on social media for product knowledge [25]. If using social media to research items makes people dependent on it and if consumers rely on reviewers’ comments on social media to obtain product information to make well-informed purchasing decisions, then we propose the following hypothesis:
**H6:** *Dependency on social media for reviewers’ comments on products influences purchase intention.*

### 1.2. Conceptual Model

This research proposed a conceptual model based on the literature, as shown in Figure 1.

The moderating role of trust and risk on the relationship between purchase intention and dependence on social media reviews is examined in this study, which establishes a precedent in this field of inquiry as previous research has examined how social media reviews affect consumers’ intentions to make purchases. Previous research has also concentrated on how trust in social media reviews influences purchase intentions. This study introduces dependency on social media into this debate. Rather than focusing solely on the association between reviews and purchase intentions, as prior research has done, the current study examines trust and risk as moderators of the relationship between dependence on social media reviews and purchase intentions.

## 2. Materials and Methods

This section provides information about the methodology employed. The current study aimed to examine how trust and risk moderate the association between purchase intention and dependence on social media reviews. The research conducted in the present study is quantitative. For this study’s results to be objective and generalizable, a significant volume of data must be gathered. Objectivity was the main reason for choosing the quantitative research method. As [42] showed in their study, quantitative studies facilitate the collection and analysis of larger datasets, which can reduce the subjectivity of research results. The details of the materials and methods are explained in the following paragraphs.

### 2.1. Participants

The participants for this study were selected from North Cyprus universities. North Cyprus universities host students from many continents and countries across the world with various cultures, ideologies, and overall perspectives on life, making the sample more diverse and representative. Compared with nonstudents, students and people in and around educational institutions have an intense online presence, which is why students were chosen as the target population. Additionally, considering how dynamic students and people in and around educational institutions are online, they are exposed to more reviewers’ comments from North Cyprus and their countries of residence.

Four of North Cyprus’s most populous universities provided the study sample. The selected schools are Near East University, which is based in Nicosia and has a population of 27,000 students [43]; Cyprus International University, based in Haspolat, which has a population of more than 22,000 students [44]; Girne American University, based in Girne, which has an 18,000-strong population of students [45]; and Eastern Mediterranean University, based in Famagusta, with an 18,000-strong population of students [46]. Table 1 shows the participants from the student population based on available information as of 2024.

The sample size for this study included 384 students selected through the conventional sampling technique. The sample sizes of previous studies that examined related topics to this study ranged between 150 and 300 [48,49]. The sample size of 200 used by previous studies does not adequately represent the number of university students in North Cyprus, so this study considered a sample size of 384, since [47] estimates the number of university students to be 108,295. More than 78% of all students in North Cyprus study at the universities selected for this study. That is,
85,000/108,295 × 100%= 78% (1)

A sample size of 384 academically advanced adults with high social media engagement was determined by employing a sample size calculator available online [50]. In addition, the 384-strong sample size is acceptable for this study, according to [51].

#### The Respondents’ Demographic Characteristics

Gender, age, and educational achievements are the three demographic attributes of the respondents that were considered for this research. An overview of the respondents’ demographics is provided in Table 2. The study’s findings indicate that 54.4% were male, 46.6% were more than 30 years old, and 58.9% had earned an undergraduate degree.

### 2.2. Data Collection Instrument

The study conducted a survey using a questionnaire developed by the researchers, using previous measurement variables and guided by theories. The survey comprised a 5-point Likert scale adapted to align with the study’s themes where 5 = Strongly Agree, 4 = Agree, 3 = Neither Agree nor Disagree, 2 = Disagree, and 1 = Strongly Disagree. The data were collected both online, through Google Forms, and in person by handing the forms directly to the participants in their schools.

After collecting resources for this research from previous related studies and settling on the aim and questions for the study, the questionnaire for this research was prepared. The questionnaire included four demographic questions. There were seven questions for variable 1 (dependency on social media for reviewer’s comments on products; action orientation scale) which was developed by [52] and the Cronbach alpha was 0.874. There were six questions for variable 2 (purchase intention) which was developed by [53] and the Cronbach alpha was 0.865. There were five questions for variable 3 (trust) which was developed by [53] and the Cronbach alpha was 0.867. There were six questions for variable 4 (seeking reviewers’ product-related comments on social media) which was developed by [54] and the Cronbach alpha was 0.779. There were four questions for variable 5 (risk) which was developed by [53] and the Cronbach alpha was 0.713. The total 28 items have a Cronbach alpha of 0.926. According to [55], Cronbach’s alpha must be at least 0.7 to obtain dependability and be considered an acceptable study. This indicated that all variables used in this research were reliable.

The convergent validity test, which is analyzed using factor loading, was used by the researchers to evaluate the study’s validity. The study also conducted a discriminant validity test.

### 2.3. Data Analysis Technique

IBM SPSS version 25 was used in this study to analyze the collected data. To identify the relationships between the observed variables, exploratory factor analysis (EFA) was used. A correlation analysis was performed to ensure that the variables were related. To test the study hypotheses, the researchers utilized SPSS version 4.2’s PROCESS macro.

## 3. Results

This study distributed 400 questionnaires to academically advanced adults with high social media engagement, who fell within the sample frame: university students at Cyprus International University, Eastern Mediterranean University, Girne American University, and Near East University. There were, however, 384 completed questionnaires. This indicates that the realization rate for the survey was 96%.

### 3.1. Factor Analysis

Researchers break down the observed variables into smaller groups and determine how they are related using exploratory factor analysis [56]. The factors were extracted using Promax with the Kaiser normalization rotation method, a principal component analysis (PCA) approach. Following [55], only items with a loading value of at least 0.4 were included in this study. Reference [57]’s specification of the requisite sample value is effectively met by the KMO of 0.903 and Bartlett’s test significance level of (P 0.05).

The results of an exploratory factor analysis revealed that there are five distinct factors that account for 61.38% of the overall variance. With seven items and a loading range of 0.469 to 0.893, DSM accounted for 34.59% of the variance in the total. The PI loaded six items between 0.591 and 0.876, and 9.81% of the overall variance was explained by this factor. Five items in the T construct, which accounted for 7.23% of the variance overall, had loadings between 0.638 and 0.880. The loadings of six items in the SRP ranged between 0.480 and 0.837, accounting for 5.01% of the variance in total. Two of the four elements that made up R’s initial construct were removed; a total of 4.74% of the variance was explained by the final two items, which loaded between 0.725 and 0.856. The results of the exploratory factor analysis are reported in Table 3.

CFA, which relies on the existence of a single dimension underpinning a set of measures, was employed to guarantee the unidimensionality of construct identification. AMOS version 24 was utilized for this purpose. Using the convergent validity test and discriminant validity tests, the researchers examined the validity of this investigation. Factor loading is one way to analyze convergent validity, according to [55]. Additionally, to attain validity, the composite reliability (CR) should be greater than or equal to 0.6, and the average variance extracted (AVE) should be 0.5 or greater. The confirmatory factor analysis results are summarized in Table 4, which also demonstrates that all the constructs satisfy the validity requirements and are dependable. 

Moreover, the authors conducted a discriminant validity test to check construct validity. According to [58], statistical tests were developed to assess discriminant validity by evaluating whether the correlation between two constructs is statistically significantly less than unity. Table 5 summarizes the discriminant validity test.

Figure 2 captures the CFA.

Furthermore, as Table 4 illustrates, [59] identified six indicators of the quality of the model fit: the comparative fit index (CFI), the normative fit index (NFI), the root mean square error of approximation (RMSEA), the incremental fit index (IFI), the standardized root mean square residual (SRMR), and the chi-square/degree of freedom (CMIN/DF). The study’s CMIN/DF value, which was 2.189, fully satisfied [60] fewer than three criteria. Additionally, the CFI, NFI, and IFI values were 0.936, 0.890, and 0.937, respectively. The criteria of [61,62,63] were all met by these indicator values, which were all very close to 0.9. Additionally, the values of RMSEA (0.056) and SRMR (0.054) met the benchmarks set by [63,64]. It is possible to conclude that the model fits the data sufficiently when taking into account the outcomes of these fitted indices (Table 6). Thus, it is possible to perform the following analysis.

### 3.2. Mean Scores of the Study Variables

The mean scores for the research variables are displayed in Table 7. For every variable, the respondents’ mean score exceeded the 3.00 midpoint level. The means scores were used to determine the degree of approval of the attitude statements and concepts in the questionnaire according to [65]. The rule specified in Table 7 is that, the agreement for the concept is strongly disagree if the average mean of the concept is between 1 and 1.79, disagree if the average mean of the concept falls between 1.8 and 2.59, neither agree nor disagree if the average mean of the concept is between 2.6 and 3.39, agree if the average mean of the concept is between 3.4 and 4.19, and strongly agree if the average mean of the concept is between 4.2 and 5. According to these results, most participants agreed that they depend on social media for product reviews; most participants agreed that social media product reviews have an impact on their decision to buy; the majority of the participants trusted reviewers’ product-related comments on social media; the majority of the participants agreed to seek reviewers’ product-related remarks on social media; and most participants agreed that negatively reviewed products are risky purchases.

Table 8 displays the correlation analysis results, which reveal that all seven constructs had a positive correlation with one another at a significance level of 0.01.

### 3.3. Hypothesis Testing

To test the study hypotheses, the researchers utilized SPSS version 4.2’s PROCESS macro. The hypothesis details are compiled in Table 9. The findings of this investigation demonstrated that the hypotheses produced statistically significant results following [66]’s criteria. The results showed that H1 (seeking reviewers’ product-related comments on social media affects purchase intentions) has a very weak relationship with the SRP and PI (*R*^2^ = 0.2201, *p* = 0.000). H2 (seeking reviewers’ product-related comments positively impacts dependency on social media) demonstrated that the SRP has a very weak impact on DSM (*R*^2^ = 0.2758, *p* = 0.000). For H3 (dependency on social media mediates the relationship between seeking reviewers’ product-related comments and purchase intentions), DSM mediates the relationship between the SRP and PI and has a moderate and positive influence on both (*R*^2^ = 0.5259, *p* = 0.000). H4 (risk moderates the relationship between seeking reviewers’ product information, dependency on social media reviews, and purchase intentions) demonstrated that R moderates the relationship between DSM and PI and has a weak impact on the relationship between DSM and PI (*R*^2^ = 0.4462, *p* = 0.070). H5 was rejected. H6 (dependency on social media for reviewers’ comments on products influences purchase intentions) indicated that DSM has a positive and very weak impact on PI (*R*^2^ = 0.2150, *p* = 0.000). Thus, except for H5, which was rejected, all of the hypotheses had statistically significant results and were accepted.

This study used Process Macro Model 14 to examine the relationships between the dependent variable (TPI), the moderator (TR), and the mediator (TDSM). According to Figure 3, as risk increases, purchase intention decreases.

## 4. Discussion

The relationship between dependence on social media reviews and purchase intentions was explored in this study to determine how risk and trust affect purchase intentions. In other words, do risk and trust make customers perceive social media reviews differently, and does the dependency on social media reviews make customers see the product or service negatively or positively? With the help of a survey of 384 academically advanced adults with high social media engagement, studying at North Cyprus universities, this study achieved its aim.

The relationship between purchase intentions and seeking reviewers’ product-related comments on social media was 22%. The results imply a weak relationship between purchase intentions and seeking reviewers’ product-related comments on social media. One reason for the poor correlation between the SRP and IP could be the excessive number of reviews. As numerous reviews and comments are available, individuals may find it difficult to sift through the data and draw a direct link between those reviews and their purchasing decisions. Ensuring that reviews are up to par can improve the overall usefulness of information [13,14,25]. This study recommends the integration of reviews with expert perspectives. This can give consumers a more balanced perspective by combining the perspectives of industry professionals and regular users.

The relationship between the SRP and DSM was 27.6%. The results indicated a weak relationship between the SRP and DSM. Social media is a useful tool for seeking information, and people become dependent on social media when they constantly seek information on it [32]; this finding has been confirmed in several studies, including that of [16]. The current study also confirms that people depend on social media for product information. The weak relationship between the SRP and DSM, however, can be explained by the lack of consensus among social media reviews, where there are differing opinions, information overload, and inconsistent information; as a result, consumers find it difficult to make educated decisions based solely on social media reviews, especially in the absence of expert opinions. The weak relationship between the SRP and DSM could also be a result of the diversity of the participants. Despite the primary sample being drawn from North Cyprus, the sample is more typical of a worldwide audience than of a homogeneous local group, bringing a variety of cultural influences, doctrines, and viewpoints. Cultural disparities, such as differing degrees of confidence in internet sources, different communication philosophies, and divergent perspectives on social media use, may nevertheless have an impact on the weak association revealed. A more comprehensive understanding of the weak relationship discovered in the study can result from these cultural elements that influence how people interpret and depend on social media reviewer’s comments. Likewise, dependency on social media depends on how well social media satisfies the need for user information [17]. Therefore, to increase the relationship between the SRP and DSM, users must find beneficial information each time they use social media. This study recommends that organizations indicate reviews from customers who have bought and utilized their products. This can solve the effect of inconsistent information that may deter the use of social media. Additionally, the availability of expert opinions can curb the effect of information overload.

DSM mediates the relationship between the SRP and PI by 52.3%. The results imply that DSM is an effective mediator in the relationship between the SRP and PI. Marketing and advertising campaigns may influence the association between the SRP and the PI. The mediating effect of DSM may be mitigated by powerful and widespread marketing messaging. Similarly, there could be a moderate mediating effect if consumers regard social media reviews as just one information source among many. The degree to which social media content is perceived as more dependable and of higher quality than information from other sources may affect the effectiveness of such mediation. Organizations must collaborate with influencers and industry experts to create credible social media reviews. Having well-known people involved might increase the impact of comments made on social media and make users more reliant on the site. The study also recommends including components of social proof in the review display, including the number of views, likes, and shares. Users’ perceptions of social media remarks can be improved by social evidence, which makes them seem more valuable.

The interaction between SRP, DSM, R, and PI was 44.5%. This implies that there was a weak relationship between the SRP, DSM, and PI, with R as the moderator. R in this study’s analysis consisted of the following attitude statements: “I would face negative consequences if I use this product/brand because of social or environmental harm”, which addressed sustainability issues, and “using the product/brand would damage my reputation or image as a person”, which addresses societal concern and identity and the need to maintain a good reputation in society. The weak relationship can therefore be attributed to users’ worries about sustainability issues, societal concerns and identity, and reputational issues. Reference [34] explained that consumers perceive a product to be risky if a review provides negative information about that product. Therefore, the researchers recommend that organizations monitor social media reviews and address any issues raised, especially issues regarding sustainability and reputational damage. Negative reviews about products should also be discussed extensively to clarify the doubts of potential customers who may depend on reviews to make a purchase decision. Resolving problems related to perceived risks can build DSM.

The relationship between dependency on social media for reviewers’ comments on products and purchase intentions was 21.5%. This implies that people are dependent on social media to make purchase decisions. Based on the results of this study, dependency on social media significantly changes the relationship between seeking reviewers’ product-related comments on social media and purchase intent. According to the previous literature and the findings of this study, it is clear that as people continuously seek information from social media, they become dependent upon it for reviews [16,18]. People will continue to seek information on social media if they find important or interesting information. This implies that if people find essential product information on social media, they will continue to seek important information on social media, and hence, they will become dependent on it, which will influence purchase intentions. This analogy could explain why the dependency on social media in this study did not improve the relationship between the SRP and PI. Thus, if people are unable to find engaging product information on social media regarding a product they wish to purchase, they may cease seeking information this way; thus, they may not be dependent on social media.

Similarly, if the information is engaging and leads to dependency on social media but is not useful for analyzing the worth of a product, it may not be able to influence a purchase. The researchers believe that the information available must be engaging and useful for DSM to mediate the SRP and PI effectively. As a result, it is recommended that reviews about products are designed to address the pertinent concerns most customers have about the product. This would require organizations to actively involve themselves in review sites, provide information, clarify misconceptions, and correct product or service mistakes. This can be achieved with social media listening tools.

### 4.1. Theoretical Implications

There are theoretical implications of this study. The media systems dependency theory and the social influence theory guided this research. Based on the results of this study, it has been demonstrated that risk moderates the reliance on social media comments and their influence on purchase intentions, which is consistent with and expands the media systems dependency theory. This emphasizes how context affects media effects and advances our knowledge of the complex interactions that occur between situational variables and media dependence. Thus, media dependency is not static but dynamic, and it can be influenced by situational variables such as perceived risk.

On the other hand, the results call into question the conventional wisdom that regards trust as a crucial element in the social influence process; trust is, after all, one of the main tenets of the relationship between an influencer and a follower. Nevertheless, the lack of relevance of trust as a moderator in the link between the intention to purchase and the desire to seek reviewers’ comments suggests the potential that risk may have a higher influence than trust, on consumer behavior, specifically, in this relationship.

### 4.2. Practical Implications

Based on the results of the study, it was found that most academically advanced adults with high social media engagement use social media to seek product-related comments from reviewers, and most depend on social media for product-related information. The influences that the SRP, DSM, and PI have on each other are significant, although their relationships are weak. Webmasters, supervisors, merchants, and social media marketers can strategically employ reviews to their advantage by making pertinent information easily accessible, offering professional reviews, and producing captivating content online. This will strengthen the correlation between the variables, thus enhancing purchase intention. 

Emphasizing sustainable practices and safeguarding brand reputation can decrease perceived risks. Managers should prioritize sustainability and reputation management in their marketing strategy, as there is a strong correlation between these aspects and risk perception. The study suggests that organizations should regularly share information about their sustainability initiatives, such as using eco-friendly materials, reducing carbon footprints, and engaging in ethical sourcing. This can be performed through social media, company websites, and press releases. Obtaining and displaying certifications from recognized environmental organizations can serve as credible endorsements of the company’s commitment to sustainability. Organizations should also develop environmentally friendly products and implement green practices in business operations, such as reducing energy consumption, minimizing environmental impact, and leveraging sustainable resources. Perceived risk is a major factor in consumers’ decision-making processes. By addressing sustainability and social reputation issues, businesses can mitigate these risks and enhance consumer confidence.

Organizations may also benefit from DSM influenced by another organization or reviewer that gave firsthand information about a product. When people find engaging information on a particular product on social media, they become dependent on social media for information on the said product. However, suppose the organization or a reviewer that provides the awareness about a product fails to answer a question or concern about the product, the potential client may purchase a similar product from another organization they find information on. Hence, completeness of information is necessary to drive sales, since different clients seek different answers, and organizations must serve the information needs of every potential client. Therefore, regardless of the source of information, whether the reviewer’s comment, reviews, advertisement, or any other source, organizations must ensure that the information provided is complete to prevent their clients from moving to substitute or related products. Likewise, an organization might increase sales of its products by providing answers to questions and addressing concerns about the products of other organizations.

Additionally, regarding managerial implications, the digital landscape is evolving, which has affected how consumers seek information on social media. With the use of artificial intelligence (AI), most organizations are providing tools that make it easier and more effective for consumers to find their product or service information [67]. For an organization to thrive, increase, or maintain its market share, it is essential to leverage AI tools and keep up with the quickly changing digital scene [67,68]. AI-powered social listening tools can be used by organizations to retrieve customer reviewers’ comments that can aid in sentiment analysis. Managing a brand’s reputation effectively through these AI-powered tools enables companies to address any unfavorable comments or concerns promptly and to maintain a positive reputation among the general public.

### 4.3. Limitations and Directions for Future Research

This study has limitations. First, the researchers acknowledge the use of a restricted sample size and therefore suggest that future researchers consider a broader sample size. Moreover, a wider sample frame that clusters people based on their culture, countries of origin, gender, or other differentiating factors is encouraged, since it would aid in comparing the attitude of different groups toward social media reviewers’ comments and purchase intention. There can be variations in the ways that people acquire and use technology, which can impact how people use social media and, in turn, how social media influences them. Another limitation of this study is that there may be confounding variables not accounted for: the results may be distorted by additional variables that affect purchase intention but were not taken into consideration. These variables could include offline influences, personal preferences, past experiences, or economic circumstances.

This study provides other recommendations for future researchers. It concluded that DSM mediates the SRP and PI. However, different moderators can significantly improve or reduce purchase intentions, even in the presence of DSM. According to the study, risk reduces PI, and trust is insignificant. If risk reduces purchase intention and trust is insignificant in the relationship, then what increases the mediating effect of DSM on the relationship between the SRP and PI? This study, therefore, recommends that future researchers conduct extensive research on factors that improve dependency on social media for product information that could influence purchase intentions. In light of this study’s findings, it would be useful to conduct a qualitative study to gather additional information. The findings of the current research recommend that future researchers introduce “user engagement” as a moderator of the relationship between the SRP and DSM and “usefulness of information” as a moderator between DSM and PI. Another area of research that could build on the findings of this study would be to examine people’s opinions about artificial intelligence (AI) and whether they are open to using AI-powered tools to obtain product-related information from social media sites.

## 5. Conclusions

In this study, we explored how reviews and people’s dependency on social media for product information contribute to purchase intentions while also considering how risk and trust moderate the relationship between dependency on social media and purchase intent. There was a gap in research regarding the subtle influence of individual reviewers’ comments as opposed to general social media reviews. As such, this study aimed to fill in the gap by specifically addressing individual reviewer’s comments. This study succeeded in filling these research gaps. This study also introduced an additional variable (DSM) in the relationship between S and PI that has not been extensively researched in previous studies. The focus of earlier research on social media dependence was on people’s addictions to platforms rather than their reliance on them for product information. Conversely, the current research highlighted social media dependence as the reliance on social media, rather than an addiction to social media.

The question guiding the study was “Do trust and perceived risk of reviewer comments on social media moderate the relationship between consumers’ dependence on social media for product information and their subsequent purchase intentions?” At the end of the study, it was concluded that trust is insignificant as a moderator in the relationship between consumers’ dependence on social media for product information and their subsequent purchase intentions. However, the perceived risk of reviewer comments on social media moderates the relationship between consumers’ dependence on social media for product information and their subsequent purchase intentions.

According to the results of this study, negatively reviewed products are risky purchases. As such, the presence of risk reduces purchase intention. Risk in this research was first defined as the fear of a product not meeting the expectations or needs of the customer, a product not being sustainable, or a product that can cause reputational damage to the user. After the analysis of the study, however, risk was redefined as the fear of a product not being sustainable and a product that can cause reputational damage to the user. The reason the study researchers redefined risk after the data analysis was because the first two items in the survey questionnaire that addressed risk as the fear of a product not meeting the expectations or needs of the customer were deleted after the initial factor loading since they were deemed irrelevant to the study. The remaining items that were considered relevant to the study addressed risk as the fear of a product not being sustainable and a product that can cause reputational damage to the user. 

As the study results revealed, people avoid products that have been negatively reviewed. What does an organization do if its products are negatively reviewed on social media? The researchers recommend that organizations that face negative reviews on social media be strategic in handling them. This may include addressing the source of concern for the reviewer. If a product is flawed, the company must fix it and respond to the review with a revised version. The goal is to turn the majority of all negative reviews in favor of the organization. Organizations may need to go the extra mile to address harmful reviews on their social media pages; this must be carried out with caution because, if not adequately handled, it may aggravate the organization’s dented image. This is the reason most organizations avoid addressing negative reviews. However, organizations that strategically address negative reviews may salvage their images.

## Figures and Tables

**Figure 1 behavsci-14-00552-f001:**
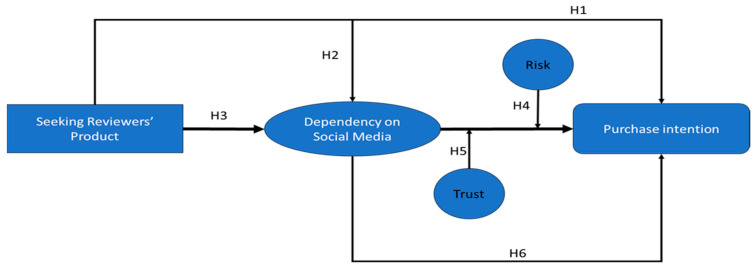
Conceptual model.

**Figure 2 behavsci-14-00552-f002:**
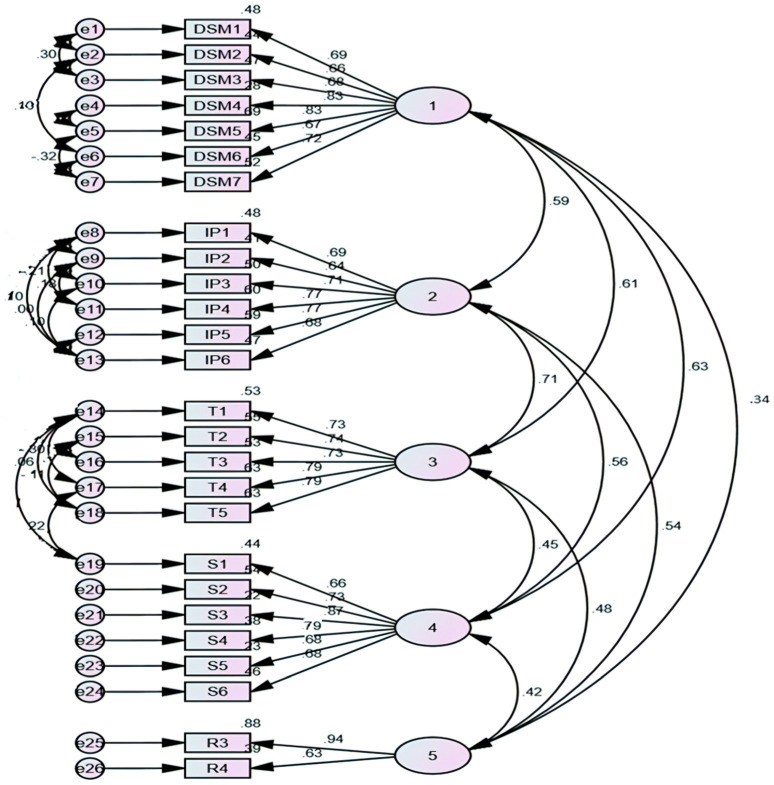
Confirmatory factor analysis.

**Figure 3 behavsci-14-00552-f003:**
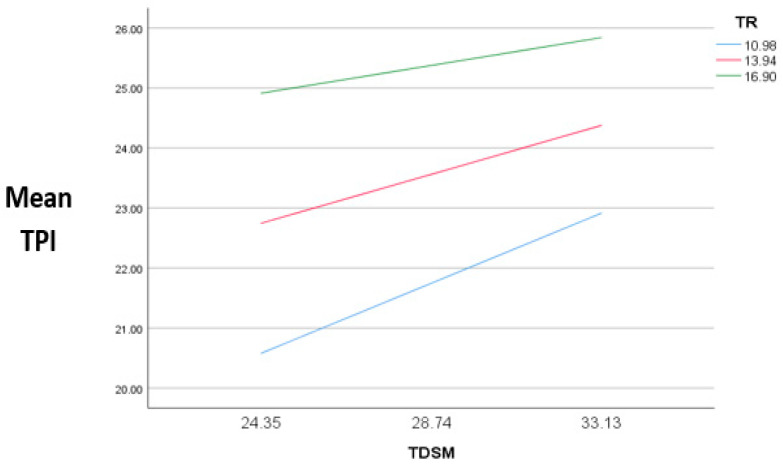
Model representation between the variables.

**Table 1 behavsci-14-00552-t001:** Population of students from which the participants were drawn.

Name of Institution	Location of School	Population of Students	Source of Information
Near East University	Nicosia, North Cyprus	27,000	[43]
Cyprus International University	Haspolat, North Cyprus	22,000	[44]
Girne American University	Girne, North Cyprus	18,000	[45]
Eastern Mediterranean University	Famagusta, North Cyprus	18,000	[46]
Total population of students in selected institutions		85,000	
Total population of students in North Cyprus		108,295	[47]

**Table 2 behavsci-14-00552-t002:** Respondents’ demographic characteristics.

**Gender**			
		**Frequency**	**Percent**
Valid	Male	209	54.4
	Female	175	45.6
	Total	384	100.0
**Age**			
		**Frequency**	**Percent**
Valid	18–24	91	23.7
	25–30	114	29.7
	30+	179	46.6
	Total	384	100.0
**Education**			
		**Frequency**	**Percent**
Valid	Undergraduate	226	58.9
	Graduate	158	41.1
	Total	384	100.0

**Table 3 behavsci-14-00552-t003:** Exploratory factor analysis results.

Factor	Factor Loading	% of Variance Explained	Initial Eigenvalues	Cronbach’s Alpha
**Factor 1: Dependency on Social Media**				
DSM1	0.837	34.59%	8.992	0.874
DSM2	0.816			
DSM3	0.893			
DSM4	0.469			
DSM5	0.724			
DSM6	0.796			
DSM7	0.564			
**Factor 2: Purchase Intention**				
PI1	0.591	9.81%	2.549	0.865
PI2	0.876			
PI3	0.802			
PI4	0.679			
PI5	0.758			
PI6	0.621			
**Factor 3: Trust**				
T1	0.880	7.23%	1.880	0.867
T2	0.808			
T3	0.839			
T4	0.638			
T5	0.802			
**Factor 4: Seeking Reviewers’ Product**				
SRP1	0.776	5.01%	1.303	0.779
SRP2	0.837			
SRP3	0.560			
SRP4	0.674			
SRP5	0.480			
SRP6	0.633			
**Factor 5: Risk**				
R3	0.725	4.74%	1.233	0.737
R4	0.856			

Note: DSM (dependency on social media), PI (purchase intention), T (trust), SRP (seeking reviewers’ product-related information), R (risk).

**Table 4 behavsci-14-00552-t004:** Confirmatory factor analysis results.

Construct	Items	Factor Loading	CR	AVE
**DSM (Factor 1)**	7	0.660–0.833	0.888	0.53
**PI (Factor 2)**	6	0.641–0.775	0.860	0.51
**T (Factor 3)**	5	0.730–0.792	0.870	0.57
**SRP (Factor 4)**	6	0.662–0.870	0.878	0.55
**R (Factor 5)**	2	0.626–0.937	0.770	0.63

Note: DSM (dependency on social media), PI (purchase intention), T (trust), SRP (seeking reviewers’ product-related information), R (risk).

**Table 5 behavsci-14-00552-t005:** Discriminant validity test.

	SRP	DSM	R	T	PI
**SRP**	**0.678**	0.525 **	0.415 **	0.381 **	0.469 **
**DSM**	0.525 **	**0.737**	0.389 **	0.502 **	0.464 **
**R**	0.415 **	0.389 **	**0.803**	0.566 **	0.590 **
**T**	0.381 **	0.502 **	0.566 **	**0.740**	0.612 **
**PI**	0.469 **	0.464 **	0.590 **	0.612 **	**0.719**

**. Correlation is significant at the 0.01 level (2-tailed). Note: DSM (dependency on social media), PI (purchase intention), T (trust), SRP (seeking reviewers’ product-related information), R (risk). AVE values are bolded and placed on the diagonal.

**Table 6 behavsci-14-00552-t006:** Fitted indicators for the CFA model.

Model	CMIN	DF	P	CMIN/DF	CFI	NFI	IFI	RMSEA	SRMR
	558.251	255	0.000	2.189	0.936	0.890	0.937	0.056	0.054

**Table 7 behavsci-14-00552-t007:** Mean scores of the study variables.

Variable Name	Number of Items	Means	Standard Deviation
DSM	7	4.11	0.829
PI	6	3.90	0.965
T	5	3.89	0.810
SRP	6	3.74	0.902
R	4	3.49	0.997

Note: DSM (dependency on social media), PI (purchase intention), T (trust), SRP (seeking reviewers’ product-related information), R (risk).

**Table 8 behavsci-14-00552-t008:** Correlations between variables.

	TSRP	TDSM	TR	TT	TPI
TSRP	1				
TDSM	0.525 **	1			
TR	0.415 **	0.389 **	1		
TT	0.361 **	0.492 **	0.576 **	1	
TPI	0.469 **	0.464 **	0.590 **	0.598 **	1

N = 384. **. Correlation is significant at the 0.01 level (2-tailed). Note: TSRP (total seeking reviewers’ product-related information), TDSM (total dependency on social media), TR (total risk), TT (total trust), TPI (total purchase intention).

**Table 9 behavsci-14-00552-t009:** Results of hypothesis testing.

	Linkage	*R* ^2^	F Test	*p*-Value	B Coefficient	Hypotheses Acceptance
**H_1_**	SRP - PI	0.2201	107.7801	0.0000	0.5640	Accepted
**H_2_**	SRP - DSM	0.2758	145.4959	0.0000	0.6174	Accepted
**H_3_**	SRP - DSM - PI	0.5259	176.0709	0.0000	0.3072	Accepted
**H_4_**	SRP - DSM - R - PI	0.4462	76.3477	0.0070	−0.0271	Accepted
**H_5_**	SRP - DSM - T - PI	0.4446	75.8512	0.0545	−0.0191	Rejected
**H_6_**	DSM - PI	0.2150	104.718	0.0000	0.474	Accepted

**Note:** DSM (dependency on social media), PI (purchase intention), T (trust), SRP (seeking reviewers’ product-related information), R (risk).

## Data Availability

The data presented in this study are available on request from the corresponding author.

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
