# Peer review of "Navigating Purchase Intentions: The Influence of Reviewers’ Comments Moderated by Risk and Trust"

_behavsci, 2024, doi:10.3390/bs14070552_

Round 1

Reviewer 1 Report

Comments and Suggestions for Authors

1) The paper mentions a gap in understanding reviewers' product-related comments and their impact on purchase intentions. However, it doesn't clearly specify what previous research has covered or where exactly the gap lies. Providing more context on existing studies and precisely outlining the gap would strengthen the paper's foundation.

2) The study model's development is not proceeding correctly. The authors should first discuss the more relevant theories in the field, identify the gap, and then develop the study model based on this gap. It's essential for the authors to thoroughly discuss all related theories in the field that serve as the foundation for developing the study model.

3) Theoretical and practical implications are missing.

While the paper briefly mentions the impact of risk on the model, it doesn't delve deeply into the implications of this finding or discuss potential limitations of the study. A more robust discussion of the results, including their practical implications for marketers and future research directions, would add value to the paper.

4)           Limitations and suggestions for future research are missing.

5) The conclusion section should include the answers to the research questions.

Reviewer 2 Report

Comments and Suggestions for Authors

Congratulations to the authors for a very well conceptualized and well-written paper.

The literature review is very good and the methodology is solid.  I am curious as to why SPSS and AMOS were used?  It seems to me that you could have just used AMOS to analyze the model (although SPSS could be used for factor analysis, I really don't see why it was used to test the hypotheses, since AMOS could have easily done so).  This is my main question for the authors - why not use AMOS to test the hypotheses?

Very strong and relevant ending with recommendations to use AI.  Excellent paper! 

Reviewer 3 Report

Comments and Suggestions for Authors

Dear Authors,

  Thank you for your interesting paper! I found your paper clearly structured, easy to follow. Yet, I found some disparities in your research.

Please find below my suggestions.

  • A brief summary

The aim of this paper is to examine how risks and trust moderate the relationship between dependency on social media for reviewers’ comments and purchase intentions. The main contribution of the paper is given by results showing that people seek product-related comments on social media when seeking information about a product they might be interested in purchasing and that risk is a factor that reduces the impact of other variables in the presented model. The paper is clearly structured, yet there are some minor disparities to be further addressed. 

  • General comment

The presented relations and results are expected results. If there were important, original outcomes of your research, results that nobody expected to reveal, what would those be?

  • Specific comments

1.     I consider the title is very general, compared to the analyzed model you propose in your research. I propose to integrate in the title “moderated by risk and trust”. Also, see the recommendation in Introduction, lines 86-89, also about the title information.

·       Abstract, line 14: “The sample consisted of 384 students”. I recommend you to integrate students into a general category, such as adults with superior studies or young adults that heavily use social media.  Also, I would not refer to students as participants in this study throughout the paper but rather as a specific category of social media consumers, defined by you.

·       Abstract, line 15: “exploratory factor analysis”.  You also used confirmatory factor analysis. I recommend you to state “exploratory and confirmatory factor analysis”.

·       Abstract, line 18: “the risk factor reduced the impact of other variables”.  As a reader, I would expect to find which variables exactly are those, so I recommend you state those variables in the abstract.  

·       1. Introduction, lines 55-57: “One study that looked at this topic was [15], whose study concluded that hotels received a greater number of bookings when there were existing and positive online reviews about the hotel.” – Please rephrase this. The idea is not clearly stated.

·       1. Introduction, lines 79-84: “(…) insightful information on regional differences in consumer behavior and the decision-making process.”. I consider this statement is not in coherence with the presented results. The presented results only reveal a mediated relationship. There are no regional differences presented in the final model. Further, you state: “This research makes a distinctive contribution to our understanding of how diverse cultural and socioeconomic circumstances impact the effectiveness and reception of social media evaluations by including participants from a range of backgrounds in North Cyprus.?” Similarly, I consider this statement is not supported by your results, as the presented relationship does not reveal any impact of diverse cultural and socioeconomic circumstances. Otherwise, the reader should find different modelling processing for diverse cultural and socioeconomic circumstances, which is not the case. Also, stating that you include “participants from a range of backgrounds in North Cyprus” is not supporting for stating “diverse cultural and socioeconomic circumstances”, which would theoretically imply participants from various regions of the world with very different cultures and socioeconomic environments. Please revise this section, according to reality and your presented results.

·       1. Introduction, lines 86-89: “In view of this, the research question guiding this study is as follows: do trust and perceived risk of reviewer comments on social media moderate the relationship between consumers' dependence on social media for product information and their subsequent purchase intentions?” – This is the reason I recommend changing the title suggesting rather the relationship between dependence on social media for product information and purchase intentions than relationship between reviewers’ comments and consumer behavior. 

·       2.2. Data collection instrument, lines 280-284. Please insert the measurement scales, with composing items, for each scale you present (variables 1-5), with sources. It is very important for a reader to clearly see the selected items. Present both initial scales and final scales, as resulted after Cronbach's alpha computation.

·       3. Results. You define sample frame as “university students at Cyprus International University, Eastern Mediterranean University, Girne American University, and Near East University”. Are results conclusions related specifically to university students at the universities you mentioned? If yes, are results generalizable? If not, how would results be useful considering you only addresses students from certain universities in North Cyprus?

·       3.1. Factor analysis. Table 3. Include the items list for each factor.

·       3.2. Mean scores of the study variables. What is the purpose for mean computation? How does this result help in the context of this research? If it helps, please explain in the given context. If not, you could remove this analysis.

·       4. Discussion. Lines 409-410: “One reason for the poor correlation between the SRP and IP is the excessive number of reviews.” This is an assumption. Therefore, I recommend you reformulate the verb from is to could be.

·       4. Discussion. Lines 421-422: “The weak relationship between the SRP and DSM, however, can be explained by the lack of consensus among social media reviews.” Do you think that this unexpected weakness may be also due to cultural differences, given the fact that you only analyzed individuals from North Cyprus? If so, please describe this assumption in more depth.

·       5. Conclusion. Lines 515-516: “However, organizations that strategically address negative reviews have salvaged their images.” Do you have specific examples of specific studies revealing this conclusion?

·       5. Conclusion. Lines 534-536: “With the use of artificial intelligence (AI), most organizations are providing tools that make it easier and more effective for consumers to find their product or service information.” Do you have specific examples of specific studies revealing this conclusion?

·       5. Conclusion. Limitations of research are missing. Among other limitations, you should also refer to the restricted sample you used.

Reviewer 4 Report

Comments and Suggestions for Authors

The paper analyses the relationship between social media users and products' reviews online. It employs a quantitative approach based on data collected via online and paper surveys of students enrolled at the four (major) universities in North Cyprus, with a participation of 96%. It delves into the significant link(s) between purchase intention, trust, risk, seeking reviewers' product-related information through statistical analysis and its interpretation.

For further suggestions towards improvement, see attached file.

Comments on the Quality of English Language

Please ask another reviewer for control of the statistical calculations. I cannot confirm them.

The paper needs editing for a more elegant style.

Round 2

Reviewer 1 Report

Comments and Suggestions for Authors

1) This study tests the convergent validity. The authors must also test discriminant validity. A table should be added that demonstrates the discriminant validity using any method, for example, the Fornell-Larcker criterion.

2) Implications must be added in a separate section. Please make it into the following subsections:

Theoretical implications

Practical implications

3) Add Limitations and directions for future research in a separate section.

Round 3

Reviewer 1 Report

Comments and Suggestions for Authors

The revised version is better than before